



# The Framework for 0-D Atmospheric Modeling (F0AM) v3.1

Glenn M. Wolfe[1,2], Margaret M. Marvin[3], Sandra J. Roberts[3], Katherine R. Travis[4], and Jin Liao[2,5]

[1]Joint Center for Earth Systems Technology, University of Maryland Baltimore County, Baltimore, MD, USA
[2]Atmospheric Chemistry and Dynamics Laboratory, NASA Goddard Space Flight Center, Greenbelt, MD, USA
[3]Department of Chemistry and Biochemistry, University of Maryland, College Park, MD, USA
[4]Department of Earth and Planetary Sciences, Harvard University, Cambridge, MA, USA
[5]Universities Space Research Association, Columbia, MD, USA

*Correspondence to*: Glenn M. Wolfe (glenn.m.wolfe@nasa.gov)

**Abstract.** The Framework for 0-D Atmospheric Modeling (F0AM) is a flexible and user-friendly MATLAB-based platform
for simulation of atmospheric chemistry systems. The F0AM interface incorporates front-end configuration of observational
constraints and model setup, making it readily adaptable to simulation of photochemical chambers, Lagrangian plumes, and
steady-state or time-evolving solar cycles. Six different chemical mechanisms and three options for calculation of photolysis
frequencies are currently available. Example simulations are presented to illustrate model capabilities and, more generally,
highlight some of the advantages and challenges of 0-D box modeling.

**1 Introduction**

The zero-dimensional (0-D) box model is a fundamental tool of atmospheric chemistry. Myriad chemical and physical
processes control atmospheric composition, and 0-D models can harness this complexity to quantify production and loss of
reactive species within a chemical system. Box models are routinely used for chemical mechanism inter-comparisons
(Archibald et al., 2010; Coates and Butler, 2015; Emmerson and Evans, 2009; Knote et al., 2015), evaluation of field
observations (Li et al., 2014; Olson et al., 2006; Stone et al., 2011; Wolfe et al., 2014), and analysis of laboratory chamber
experiments (Fuchs et al., 2013; Paulot et al., 2009a).

The power of the 0-D box model stems partly from its simplicity, but this also imparts inherent limitations. Such
models do not explicitly simulate horizontal and vertical transport processes, thus boundary conditions can strongly
influence concentrations of intermediate- to long-lived species like ozone. Steady-state conditions are often assumed when
constraining with or comparing to field observations, but this assumption is invalid in some situations (e.g. near large or
variable emission sources), and the history of an air mass is not always known. Chemical rate constants and observational
constraints also carry significant uncertainties, and the best way to propagate this uncertainty through to model results is not
always clear. Thus, one should not necessarily expect a 0-D box model to get "the right answer" except in cases where the
model setup is a fair representation of the true atmosphere. Rather, a box model is a platform for gaining conceptual
understanding and testing hypotheses through targeted sensitivity simulations and comparison with observations.



There is a need for user-friendly model tools within both the experimental and modeling communities. Several models are currently freely available, including the Dynamically Simple Model for Atmospheric Chemical Complexity (DSMACC) (Emmerson and Evans, 2009), Chemistry As A Box Model Application (CAABA) (Sander et al., 2011a; Sander et al., 2005), and Box Model Extensions to KPP (BOXMOX) (Knote et al., 2015). These models are written in FORTRAN,

which is a preferred language for atmospheric computation but is not the most accessible for novice programmers. Many research groups also develop their own models for specific problems, but this can be a time-consuming and error-fraught effort.

The Framework for 0-D Atmospheric Modeling (F0AM) is a versatile and open platform for simulating atmospheric chemical systems. F0AM is unique from other community box models in several respects. First, it is written in a

10 high-level programming language. Second, it is easily adaptable to laboratory, Lagrangian, and steady state applications. Third, it incorporates a suite of common explicit and condensed chemical mechanisms used in the air quality and atmospheric chemistry communities. Here we provide a general description of F0AM architecture, demonstrate several common applications, and suggest potential future improvements. Through this discussion, we also hope to elevate community awareness of the advantages and challenges of the 0-D box modeling approach.

## 2. Model Description

Earlier versions of the F0AM architecture evolved from the 1-D Chemistry of Atmosphere-Forest Exchange (CAFE) model, which was designed to resolve physical and chemical processes within a forest canopy (Wolfe and Thornton, 2011; Wolfe et al., 2011a; Wolfe et al., 2011b). In its previous incarnation, the 0-D model was referred to as the University of Washington

Chemical Model (UWCM) and applied to a variety of research problems, including investigation of lab chamber experiments (Kaiser et al., 2014; Wolfe et al., 2012), radical production and VOC oxidation in biogenic environments (Kaiser et al., 2016; Kaiser et al., 2015; Kim et al., 2015b; Kim et al., 2013; Wolfe et al., 2014; Wolfe et al., 2015; Wolfe et al., 2016), biomass burning plumes (Busilacchio et al., 2016; Müller et al., 2016), and chlorine chemistry (Riedel et al., 2014).

The design of F0AM stems from two principles: accessibility and flexibility. Accessibility refers to the ease with which any user can run the model. F0AM is written entirely in MATLAB (developed by MathWorks). MATLAB is a higher-level language than FORTRAN and can be less computationally efficient; however, it is easier to learn for researchers with little programming experience and is used extensively by the experimental community. Though MATLAB itself is not free, F0AM is provided free to the community under the GNU general public license, does not rely on MATLAB toolbox

extensions, and is open source to the extent possible. Support materials include a detailed user manual and several example setups.



Flexibility refers to the ease with which a user can adapt the model setup to a particular research problem. Front-end options enable various features and simplify switching between parameterizations and mechanisms. All inputs and options are specified in a single script. Example setup scripts cover a range of typical modeling scenarios and can act as a starting point for new scenarios or datasets. Users should not have to modify source code except in special circumstances.

A general overview of model inputs, outputs, and parameterizations is given below. Here, a model "run" refers to a single model call, while a model "step" refers to model execution for a single set of initial meteorological and chemical conditions. There can be multiple steps within a run.

### 2.1 Observational Constraints

Required meteorological inputs include pressure, temperature, and water vapor content. Several options are available to drive
the various photolysis schemes (described further below), including direct input of observed photolysis frequencies (J-values), solar zenith angles, or an actinic flux spectrum. For each chemical species, concentrations can be initialized to observed values and are either held constant throughout a model step or allowed to evolve over the course of a step.

A special option is available to force total $NO_x$ (= NO + $NO_2$) to input values at the beginning of a step. This provides a means of replenishing $NO_x$ without perturbing the modelled NO/$NO_2$ ratio, which may be desirable e.g. for
diurnal cycles of radical chemistry. Figure 1 compares predicted and observed $NO_x$ mixing ratios for a diurnal cycle simulation using this option (see Sect. 3.2 for details). For this particular example, daytime NO is slightly over-predicted while $NO_2$ is under-predicted, which could be related to the model $NO_2$ photolysis frequency (which is not measurement-constrained). Total model $NO_x$ is lower than observations by 2 ± 4% on average. When using this option, it is preferable to keep the model step interval significantly smaller than the $NO_x$ lifetime to minimize $NO_x$ loss over the course of a step. In
this example the step interval is 15 minutes and the mid-day $NO_x$ lifetime is on the order of hours.

### 2.2 Photolysis

Photolysis frequencies control radical production and the lifetimes of numerous compounds. Accurate simulation of J-values is challenging due to the variety of factors that influence the radiation field, many of which are often unknown or require some effort to determine (e.g. surface albedo, overhead ozone column, cloud and aerosol extinction or enhancement). F0AM
provides three options for calculating J-values: bottom-up, MCM, and hybrid.

In the "bottom-up" method,  J-values are calculated by integrating the product of a user-specified actinic flux spectrum with literature-derived cross sections and quantum yields. Cross sections and quantum yields are taken from the latest IUPAC (Atkinson et al., 2004, 2006) and JPL (Sander et al., 2011b) recommendations when available, and all sources are documented in a single spreadsheet. Spectra, cross sections and quantum yields are convolved using a trapezoidal
integration scheme identical to that employed in NCAR's Tropospheric Ultraviolet and Visible radiation model (TUVv5.2, available at https://www2.acom.ucar.edu/modeling/tropospheric-ultraviolet-and-visible-tuv-radiation-model). This option is most useful when simulating photochemical chamber experiments with non-solar light sources.




The Master Chemical Mechanism (MCM) provides a trigonometric parameterization based on solar zenith angle (SZA).

$$J = I \cos(SZA)^m \exp(-n \sec(SZA)) \tag{1}$$

Here, $I$, $m$, and $n$ are constants unique to each photolysis reaction, derived from least-squares fits to J-values computed with

fixed solar spectra and literature cross section and quantum yields. As discussed in Jenkin et al. (1997) and Saunders et al. (2003), solar spectra underlying this parameterization were calculated from a two-stream radiative transfer model for clear sky conditions on July 1 at a latitude of 45° N and an altitude of 0.5 km. Cross sections and quantum yields generally follow IUPAC recommendations as documented on the MCM website (http://mcm.leeds.ac.uk/MCM/). When using this option with a chemical mechanism other than the MCM, photolysis frequencies for reactions not included in the MCM are calculated

using the "hybrid" method (below) with a fixed altitude of 0.5 km, overhead ozone column of 350 DU and surface albedo of 0.01 (see below for justification).

The "hybrid" method combines the "bottom up" cross sections and quantum yields with solar spectra derived from TUVv5.2. A total of 20,064 solar spectra were calculated offline over a range of SZA (minimum/increment/maximum of 0/5/90°), altitude (0/1/15 km), overhead ozone column (100/50/600 DU) and albedo (0/0.2/1). J-values calculated for all

solar spectra are organized into a set of lookup tables. At the start of a model run, input SZA, altitude, ozone column and albedo are used for linear interpolation across these tables. This method is a compromise between the MCM parameterization, which is efficient but incomplete and optimized for surface conditions, and running the full TUV model inline, which provides greater user control but is computationally expensive and not easily modified. Also, the hybrid method is fully traceable: cross sections and quantum yields are documented in a single file, and both TUV-derived actinic

fluxes and the code for calculating J-value lookup tables are available.

Figure 2 compares photolysis frequencies calculated with the MCM parameterization, the F0AM hybrid method, and directly from TUVv5.2 for a single set of inputs (SZA = 0°, altitude = 0.5km, albedo = 0.01, $O_3$ column = 350 DU). The overhead $O_3$ column and albedo for this comparison are chosen to optimize average agreement between the hybrid and MCM values. Hybrid and TUVv5.2 photolysis frequencies generally agree to within ±20%, as expected since both utilize the

same solar spectra and (usually) comparable cross sections and quantum yields. Differences between the TUV and hybrid values for $C_2H_5CHO$ and $CH_3COCH_3$ photolysis are due to known errors in TUVv5.2 that will be resolved in the next release (S. Madronich, personal communication, 2016). Differences for MEK, $CH_3CHO$, and $N_2O_5$ are due to the choice of quantum yields. MCM J-values are more variable with respect to hybrid J-values. This is partly due to varying quantum yields; for example, MCM uses different branching ratios for the glyoxal photolysis channels. Differences may also stem

from the radiation model used to generate the MCM parameterization, but solar spectra are not available for comparison.

Any J-value can be constrained to observations via direct input. It is also possible to specify a scaling factor for all parameterized J-values. Typically this scaling is taken as the ratio of an observed photolysis frequency to its model-calculated value. Scaling to observed values is encouraged when working with field observations, as neither the MCM or hybrid methods capture the full extent of atmospheric properties that can influence solar radiation. For example, in the steady



state simulation discussed in Section 3.4, removing constraints on J(NO$_2$) and J(O$^1$D) increases average calculated NO, OH and HCHO mixing ratios by 34%, 40% and 11%, respectively.

**2.3 Chemistry**

Table 1 lists the gas-phase chemical mechanisms currently available with F0AM. The MCM is a prevalent explicit
mechanism, and version 3.3.1 (Jenkin et al., 2015) contains numerous updates to reflect recent laboratory and theoretical advances. MCMv3.2 (Saunders et al., 2003) is included for comparison purposes. Several MCM extensions are also available, including simplified monoterpene and sesquiterpene oxidation (Wolfe and Thornton, 2011), chlorine-VOC reactions (Riedel et al., 2014), and a subset of bromine and chlorine reactions from MECCA (Sander et al., 2011a). The Carbon Bond mechanisms, CB05 (Yarwood et al., 2005) and CB6r2 (Hildebrandt Ruiz and Yarwood, 2013), and the
Regional Atmospheric Chemistry Mechanism version 2 (RACM2) (Goliff et al., 2013) are condensed mechanisms commonly used in regional air quality applications. The version of the GEOS-Chem mechanism included with F0AM is based on GEOS-Chem v9-02 (Mao et al., 2013) with updates to isoprene chemistry as described in several recent publications (Fisher et al., 2016; Kim et al., 2015a; Marais et al., 2016; Travis et al., 2016). Toggling between various mechanisms is straightforward through the setup script. None of the above mechanisms include heterogeneous or aerosol-
phase processes.

Chemical rate equations are integrated with MATLAB's ode15s solver, which is designed specifically for stiff systems. A utility is also available for converting mechanisms from the FACSIMILE (MCPA Software) format into the F0AM input format. We hope that the community will continue to add to the F0AM mechanism library and that this model can serve as a platform for inter-comparing and evaluating updates to these mechanisms.

**2.4 Dilution**

A major shortcoming of the 0-D box modeling approach is the lack of explicit representation of transport processes (entrainment, dilution, etc.), which has several practical consequences. First, primary emissions like NO$_x$ and hydrocarbons must be constrained or otherwise re-supplied to compensate for chemical loss. Emissions can also be parameterized explicitly but require knowledge of the boundary layer depth and assumed instantaneous mixing. Second, a generic "physical
loss" lifetime of 6 – 48 hours is often assigned to all species to mitigate build-up of long-lived oxidation products over multiple days of integration. Model users must be aware of the limitations imposed by these choices. For example, constraining NO$_2$ is not appropriate when investigating ozone production, and the choice of physical loss lifetime can affect simulated OH reactivity (Edwards et al., 2013; Kaiser et al., 2016).

F0AM adopts a simple parameterization for first-order ventilation:

$$\frac{d[X]}{dt} = -k_{dil}([X] - [X]_b) \tag{2}$$





Here, $[X]$ is the chemical concentration, $[X]_b$ is a fixed "background" concentration, and $k_{dil}$ is a 1$^{st}$-order dilution rate constant. Expansion of Eqn. (2) shows that this parameterization is effectively the combination of a 0$^{th}$-order source ($k_{dil}[X]_b$) and a 1$^{st}$-order sink ($-k_{dil}[X]$). The choice of $k_{dil}$ and $[X]_b$ depends on the particular problem. The dilution rate constant can be set to a constant value or parameterized using additional information, such as the decrease of conserved tracers in evolving plumes (Dillon et al., 2002; Müller et al., 2016), wind speed (Bryan et al., 2012), or boundary layer growth rate (Kaiser et al., 2016). Background concentrations are typically set to up-wind, out-of-plume, or free tropospheric values depending on the system and available information. Setting $[X]_b$ to zero yields a simple 1$^{st}$-order sink, analogous to the physical loss lifetime discussed above. Regardless of the application, it is important to justify the choice of $k_{dil}$ and $[X]_b$ and/or perform sensitivity simulations to characterize how uncertainties in physical processes impact model interpretation.

## 2.5 Execution Options

Much of the flexibility of F0AM stems from up-front control of how integration proceeds across a single step and between steps. For example, the end points of one step can be used to initialize the next step, or each step can be treated as independent. The former option is appropriate for simulating the time evolution of field observations (which may have time-varying input constraints), while the latter is useful for modeling multiple chamber experiments or performing a sensitivity study (e.g. the effect of varying levels of NO$_x$ on isoprene oxidation). A "solar cycle" option is also available to make photolysis frequencies evolve "in real time" over the course of a model step, which is a standard procedure when modeling aircraft observations (Olson et al., 2006). In this case, the user must also specify location and time. It is left to the user to determine the appropriate total integration time - no convergence criteria are incorporated into model execution.

## 2.6 Output and Analysis

Model output is collected in a single hierarchical structure and includes calculated chemical concentrations and reactions rates, as well as inputs. Outputs can include all intermediate concentrations and rates along each step or values at the end of the step only (specified during setup). Tools are also provided for manipulating and plotting output; some example plots are shown below. One tool of special note is a function to identify MCM species with specific chemical functionalities (carbonyls, nitrates, etc.) using simplified molecular input line entry system (SMILES) strings (Weininger, 1988). This tool is useful for examining groups of compounds (e.g. Fig. 4) and has been used previously to develop a rough deposition parameterization for many MCM species (Kaiser et al., 2016).

## 3. Example Applications

Here we describe several common applications and demonstrate typical methods for analysis of model output. Model setup files and input data for all examples described here are included with the F0AM distribution.



## 3.1 Photochemical Chamber

Photochemical chambers are a standard tool for isolating and characterizing chemical processes. 0-D models are useful for both planning experiments and interpreting data (e.g. by testing proposed mechanism modifications). Here, we use F0AM with MCMv3.3.1 to predict $NO_x$-dependent yields of several isoprene oxidation products. For these simulations, model

meteorology is set to nominal values (298 K, 1000 mbar, 10% RH). J-values are calculated using the "bottom-up" method with a light spectrum corresponding to UV bulbs with output centered at 350 nm (Crounse et al., 2011). The model is initialized with 10 ppb of isoprene, 200 ppb of hydrogen peroxide (a common OH source) and $NO_2$ mixing ratios ranging from 10 ppt to 10 ppb. The model is integrated 1 hour for each initial $NO_2$ concentration, and yields are calculated as the slope of product gained against isoprene lost over minutes 10-15 (see inset in Fig. 3).

Figure 3 shows the yields of three first-generation products that track the fate of isoprene hydroxyperoxy radicals ($ISOPO_2$): methyl vinyl ketone (MVK) and methacrolein (MACR) from the NO channel, isoprene hydroxyhydroperoxides (ISOPOOH) from the $HO_2$ channel, and hydroperoxyaldehydes (HPALD) from unimolecular isomerization. The chemistry shifts from $HO_2$ to NO-dominated at 0.2 ppb of initial $NO_2$. Such plots can help define optimal experiment conditions and strengthen intuition regarding expected relationships in both the laboratory and the real atmosphere.

## 15 3.2 Lagrangian Plume Evolution

The time evolution of a plume – from a wildfire, urban core, power plant, or other strong emitter – offers a natural experiment for testing chemical understanding. As an example, we simulate a young biomass burning plume sampled from an aircraft during NASA's DISCOVER-AQ mission (Deriving Information on Surface Conditions from Column and Vertically Resolved Observations Relevant to Air Quality, data available at DOI 10.5067/Aircraft/DISCOVER-AQ/Aerosol-

20 TraceGas). Plume sampling occurred longitudinally from the source to ~13.5km downwind, corresponding to a processing time of ~1 hour. Model setup is identical to that described in Müller et al. (2016). Briefly, gas concentrations are initialized with mixing ratios observed over the first 1 km and include $O_3$, CO, $CH_4$, $NO_x$, HONO, and a suite of 17 reactive VOC. All gas concentrations are allowed to evolve freely in time. Meteorological conditions are updated every 250 seconds (roughly every 1 km). The dilution constant is calculated using the observed decay of CO; the dilution lifetime ($1/k_{dil}$) increases from

25 6 min to 106 min over the simulation period. Background concentrations are taken from measurements outside the plume. Chemistry is MCMv3.3.1 using MCM's default photolysis scheme, with additional reactions for initial oxidation of furfural and furan.

      Figure 4 illustrates the simulated progression of total oxidized nitrogen ($NO_y$). $NO_x$ decreases by over a factor of 2 over the course of an hour, but this is mostly balanced by formation of peroxy nitrates (mainly peroxyacetyl nitrate, PAN)

and nitric acid. As presented in Müller et al. (2016), the model quantitatively replicates the observed conversion of $NO_x$ to PAN, as well as the formation of ~60 ppb of $O_3$. The excellent model-measurement agreement for this case suggests that more advanced frameworks that account for Gaussian dispersion (Alvarado and Prinn, 2009) may not always be necessary,





but this likely depends on the nature of each case study and available constraints. On the other hand, the model does not capture the increase in some oxidized VOC, such as formaldehyde, likely indicating some missing VOC precursors. In conjunction with a detailed dataset, a box model can help to characterize the nature of such "missing" reactants and quantify the impact of these compounds on downwind chemistry.

## 3.3 Boundary Layer Diurnal Cycle

Ground-based field intensives can provide detailed data sets for driving model simulations. Here we use a subset of observations from the 2013 Southeast Oxidants and Aerosol Study (SOAS, data available at http://www.eol.ucar.edu/field_projects/sas). Observations from the Centreville, Alabama site are averaged over the entire campaign to a diurnal cycle in 1-hour intervals. There is substantial day-to-day variability in this dataset, and this coarse averaging procedure is for illustrative purposes only. Chemical constraints include $NO_x$, OH, CO, PAN, and a suite of ~35 volatile organic compounds. Total $NO_x$ is semi-constrained using the "fixed $NO_x$" option (Fig. 1), and to facilitate this we interpolate the hourly-averaged data to a 15-minute time-base. Ozone is initialized for the first step only. We use MCMv3.3.1 chemistry and the hybrid J-value parameterization with a fixed $O_3$ column of 320 DU and albedo of 0.05, without further scaling (no radiation measurements are available). A physical loss lifetime of 24 hours (using the dilution parameterization) is applied to all species. The model run extends over 4 days, using the same constraints for each day.

Figure 5(a) shows the evolution of ozone over the four-day simulation period. Ozone is in near-steady state by the end of the fourth day; concentrations increase by less than 2% between days 3 and 4. Ozone growth is rapid in the morning but slows around noon, concomitant with reduced $NO_x$ (Fig. 1). The dominant fate of organic peroxy radicals also shifts from reaction with NO to reaction with $HO_2$ at this time (Fig. 5(b)), which likely also contributes to reduced ozone production (less radical cycling) and may impact production of aerosol precursors, such as epoxides (Paulot et al., 2009b). Through sensitivity simulations that probe the timing of such changes, box modeling facilitates rapid-fire testing of multiple hypotheses and full leveraging of comprehensive datasets.

Despite good model-measurement agreement for ozone mixing ratios in the afternoon, there are significant discrepancies. Between hours 7 – 12, observed ozone increases by 23 ppb while modelled values only increase by 15 ppb. This is likely due to a lack of residual layer entrainment in the model, which can be a significant ozone source in the morning (Su et al., 2016). The model also under-predicts the evening ozone decay rate by a factor of 2, potentially implying inadequate treatment of deposition (dilution is the only physical loss in our setup). These issues highlight some of the challenges of simulating near-surface composition in a complex environment with a relatively simple model. Additional functionality may be added in the future to better represent physical processes.

## 3.4 Mechanism Inter-comparison

Regional and global models employ a variety of chemical mechanisms. Box models can isolate the chemistry contribution to inter-model differences and pinpoint potential shortcomings in condensed mechanisms. Here we show an example





comparison between all mechanisms included in F0AM (Table 1). Constraints are taken from airborne observations acquired in the Atlanta area during the 12 June 2013 flight of the Southeast Nexus mission (SENEX, data available at http://esrl.noaa.gov/csd/groups/csd7/measurements/2013senex/P3/DataDownload/) (Warneke et al., 2016). Figure 6(a) shows time series of altitude, $NO_x$, and isoprene mixing ratios for the representative flight segment, which includes

(sequentially) a vertical profile, a boundary layer transect downwind of a power plant plume, and a pass through the Atlanta urban core. Chemical constraints include 1-minute average observations of $CH_4$, CO, $O_3$, $NO_2$, PAN, methanol, and isoprene. Hybrid J-values are corrected by the average ratio of observed-to-calculated $J(NO_2)$ and $J(O^1D)$. For each 1-minute interval, the model is run with a 1-hour time step for 5 days in "solar cycle" mode to achieve steady state.

Figures 6(b), (c) and (d) compare modelled OH, $HO_2$, and OH reactivity (inverse OH lifetime) for all mechanisms.

OH concentrations agree to within ±30%, and $HO_2$ concentrations and OH reactivity to within 20%, over the whole period. The most obvious discrepancy is the somewhat low values of $HO_2$ for RACM2. To investigate further, we can compare rates of $HO_2$ production and loss between RACM2 and MCMv3.3.1. $HO_2$ lifetimes of $10 - 50$ s are nearly identical for both mechanisms, thus the difference must be related to production. Figure 7 compares $HO_2$ sources for the two mechanisms. The production of $HO_2$ from OH reactions with HCHO, CO, and other compounds is significantly slower in RACM2. RACM2

OH concentrations, however, fall in the middle of the pack. Furthermore, that agreement is much better in the high-altitude portion, where isoprene is absent. Taken together, these results suggest minor issues in the distribution of isoprene oxidation products in RACM2. The utility of direct rate analysis afforded by box models cannot be overstated, especially for chemical species with multiple sources and sinks.

## 4. Future Functionality

F0AM is a community tool that will continue to evolve. A range of modifications are envisioned to improve functionality, including:

- Propagation of uncertainties in constraints and rate constants, e.g. using Monte Carlo methods
- Explicit deposition and emission parameterizations
- Gas-particle partitioning and heterogeneous chemistry
- Lagrangian trajectory model interface
- Tagging of oxidation products for source apportionment

Development of these capabilities will be driven by the specific requirements of new modeling projects.



## 5. Code Availability

F0AM is available for download at https://sites.google.com/site/wolfegm/models. Version 3.1 is included as a supplement to this publication. Frequent users are also encouraged to join the F0AMusers@googlegroups.com mailing list/forum and to share newly-developed code between the community.

## Acknowledgements

We are grateful to Kirk Ullmann for providing the TUV model executable and Markus Müller for providing the setup and data for the Lagrangian plume example. Photolysis parameterizations are based on code developed by John Crounse, Fabian Paulot and Wyatt Merrill. Jen Kaiser provided helpful feedback on the model documentation. Emma D'Ambro provided helpful comments on the manuscript. We are also indebted to the many scientists and crew members of the DISCOVER-AQ, SENEX and SOAS field missions for collecting observations used to constrain the example simulations. GMW acknowledges support from the NOAA Climate and Global Change Postdoctoral Fellowship Program and NASA ACCDAM grant NNX14AP48G. MRM acknowledges support from a NASA Earth Systems Science Fellowship.

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



**Table 1. Chemical Mechanisms in F0AM.**

| Mechanism | # of Species | # of Reactions | Reference |
|---|---|---|---|
| **MCM v3.3.1** | 610[a]<br>5832[b] | 1974[a]<br>17224[b] | Jenkin et al. (2015) |
| **MCM v3.2** | 455[a]<br>5734[b] | 1476[a]<br>16940[b] | Saunders et al. (2003) |
| **CB05** | 53 | 156 | Yarwood et al. (2005) |
| **CB6r2** | 77 | 216 | Hildebrandt Ruiz and Yarwood (2013) |
| **RACM2** | 124 | 363 | Goliff et al. (2013) |
| **GEOS-Chem** | 171 | 505 | Mao et al. (2013); Marais et al. (2016); Fisher et al. (2016); Travis et al. (2016); Kim et al. (2015a) |

[a]Isoprene, methane and inorganic reactions only. [b]Full mechanism.





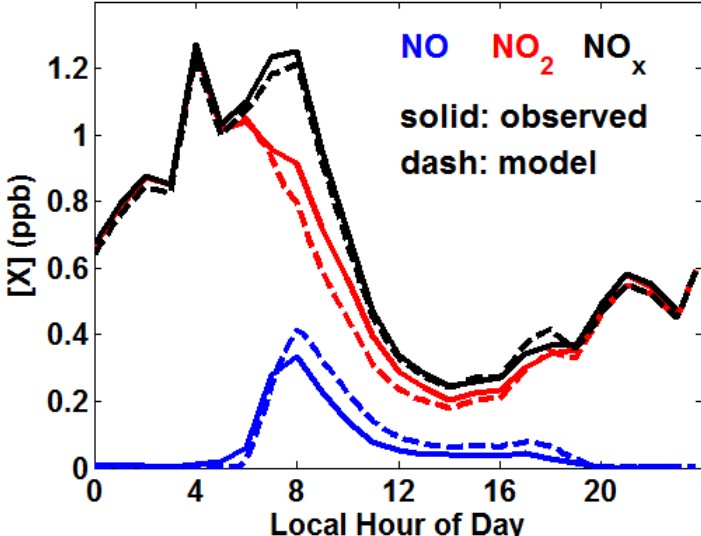

**Figure 1.** Comparison of simulated (dashed) and observed (solid) mixing ratios of NO (blue), $NO_2$ (red) and $NO_x$ (black) for the diurnal cycle setup described in Sect. 3.2. This simulation uses the "fix $NO_x$" option, which resets total $NO_x$ to the observed value at the start of every step (15 minutes, in this case) while maintaining the calculated $NO/NO_2$ ratio.





**Figure 2. Ratio of photolysis frequencies calculated from the MCMv3.3.1 SZA parameterization (red triangles) and TUVv5.2 (blue circles) against the F0AM hybrid method. Ratios are taken for J-values calculated with a single set of inputs (SZA = 0°, altitude = 0.5 km, albedo = 0.01, O$_3$ column = 350 DU). Blue and red numbers denote values falling outside the y-axis range. All reactions in the hybrid scheme are shown, though some do not have TUV or MCM analogues.**





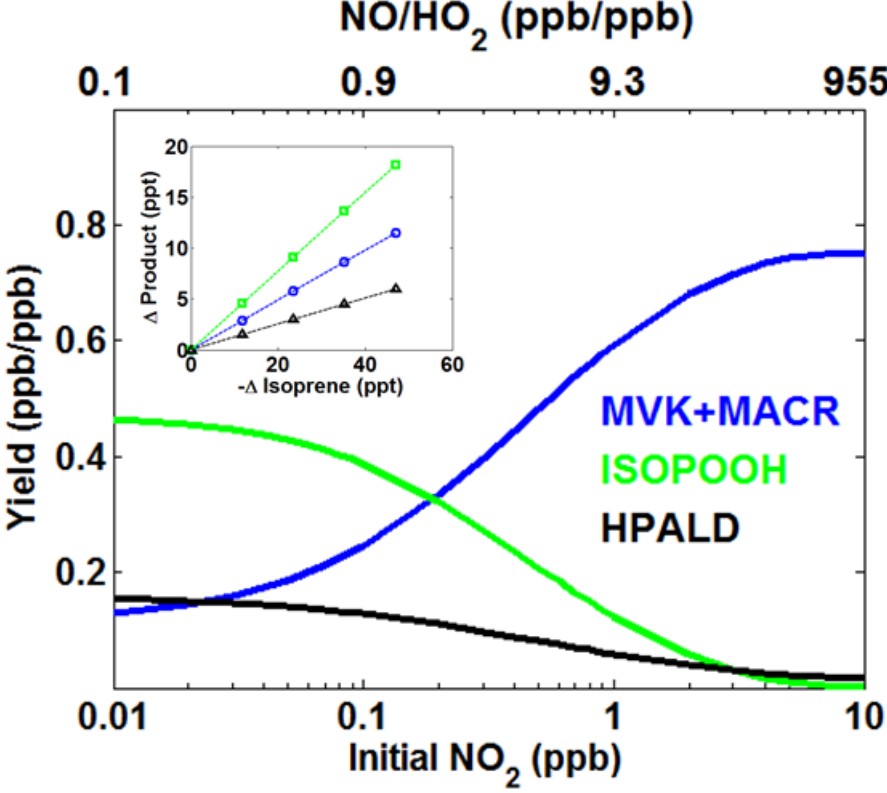

**Figure 3. Theoretical yields of first-generation isoprene oxidation products for a series of isoprene oxidation experiments with varying levels of NO$_x$ (Sect. 3.1). Yields are calculated as the slope of product formed versus isoprene lost over minutes 10-15 of oxidation (example shown in inset). The upper axis shows average NO/HO$_2$ ratios over the same period.**



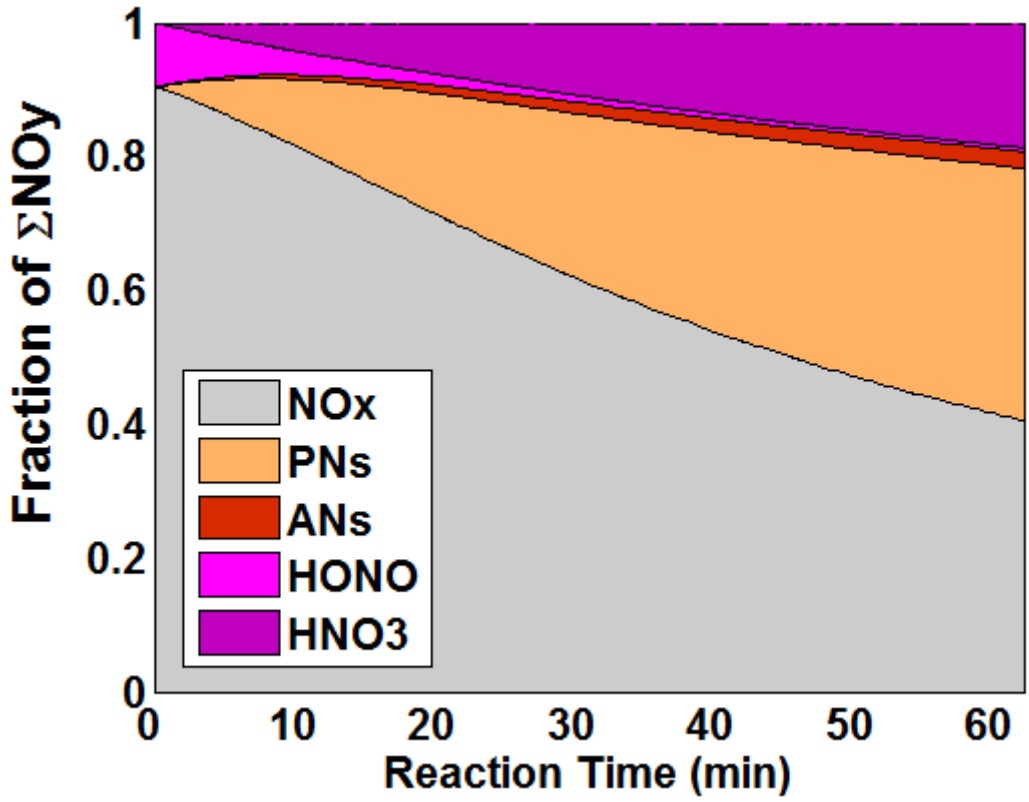

**Figure 4. Simulated evolution of total oxidized nitrogen in a nascent biomass burning plume as described in Sect. 3.2. "PNs" represents all peroxy nitrates, and "ANs" represents all alkyl nitrates. The PNs and ANs groups were generated using an algorithm that scans MCM SMILES strings (see Sect. 2.6).**



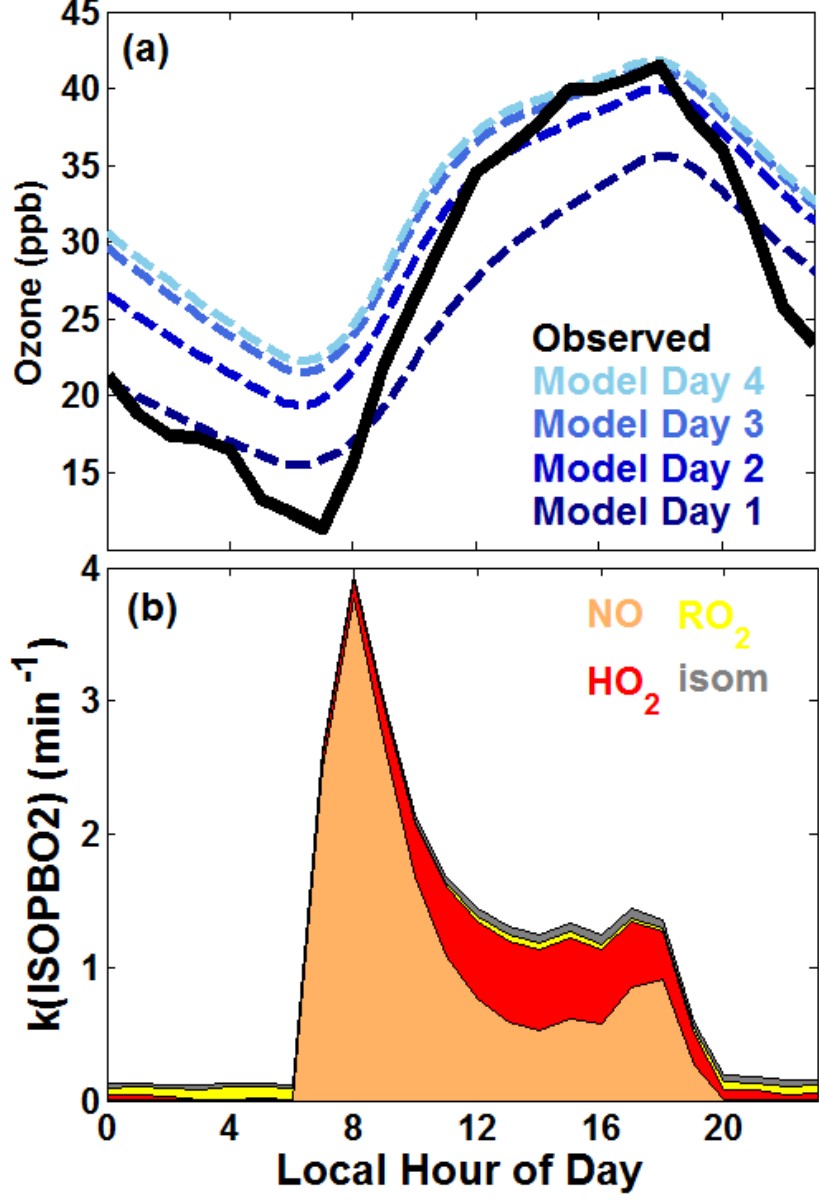

**Figure 5.** (a) Progression of simulated diurnal ozone profile (dashed lines) over four days of a constrained boundary layer diurnal cycle simulation (Sect. 3.3). Observed ozone is also shown (solid black line). (b) Reactivity of a representative first-generation isoprene hydroxyperoxy radical against reaction with NO (orange), $HO_2$ (red), other $RO_2$ (yellow) and 1,5 H-shift isomerization. Rates are taken from the final simulation day.





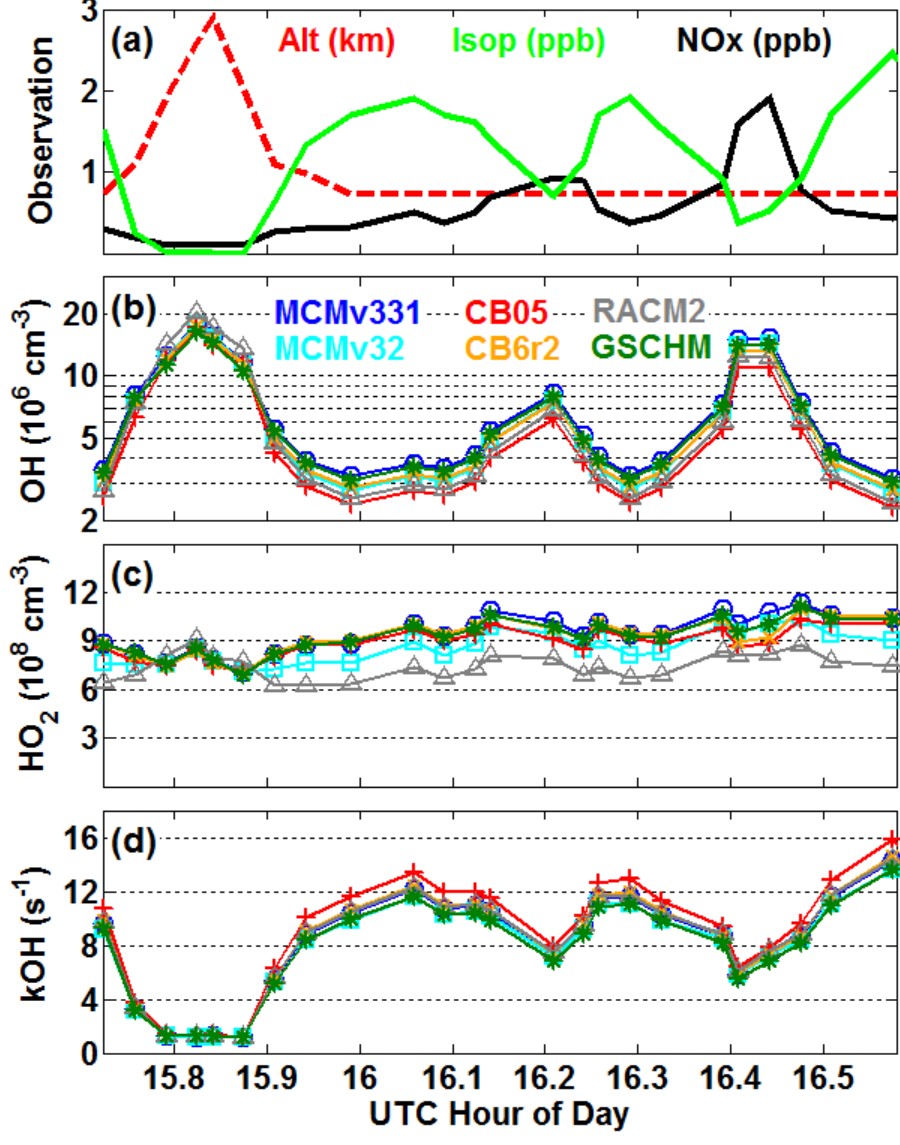

**Figure 6. (a) Time series of pressure altitude (red dashed line) and observed mixing ratios of isoprene (green line) and NO$_x$ (black line) for the SENEX Atlanta area flight leg discussed in Sect. 3.4. Observations from this dataset drive steady-state simulations for comparison of modelled OH (b), HO$_2$ (c) and OH reactivity (d) among six chemical mechanisms: MCMv3.3.1 (blue circles), MCMv3.2 (cyan squares), CB05 (red +), CB6r2 (orange x), RACM2 (gray triangles) and GEOS-Chem (green asterisks).**




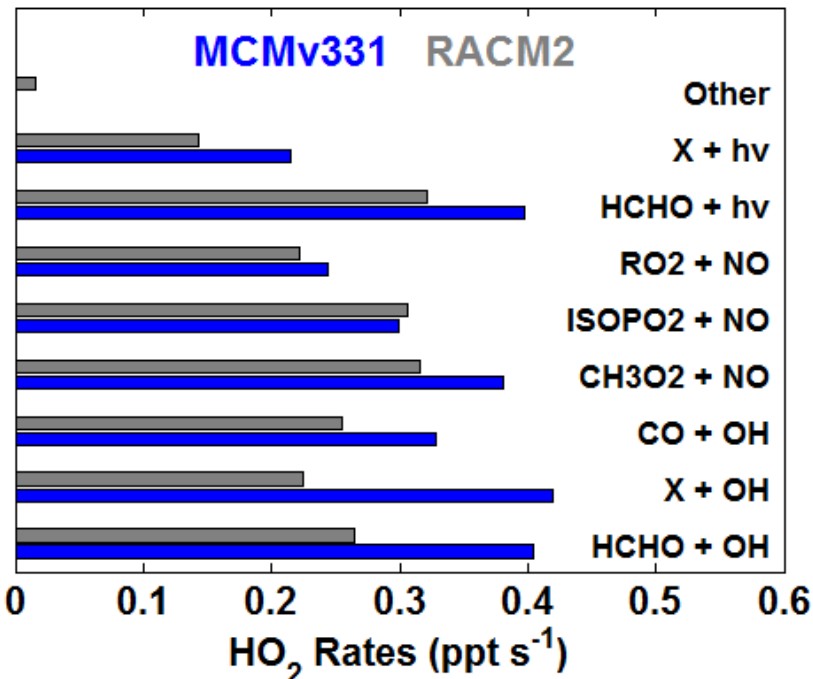

**Figure 7. Comparison of HO₂ sources for the MCMv331 (blue) and RACM2 (gray) steady state simulations. Production rates are instantaneous values from the model step at UTC hour 16.2 (see Fig. 6). In the labels, "X" and "RO2" refer to all HO₂-producing species other than those listed explicitly.**