# Peer review of "The Framework for 0-D Atmospheric Modeling (F0AM) v3.1"

_Geoscientific Model Development, 2016_

## Referee Comment (RC1) · Anonymous Referee #1 · 15 Jul 2016

This manuscript introduces the Framework for 0-D Atmospheric Modeling (F0AM) v3.1, a MATLAB-based platform that emphasizes accessibility and flexibility. This framework has already proven to be a useful tool implemented in previous studies, and with the modifications and added functionality detailed in the manuscript, I anticipate an increase in use and development from the atmospheric modeling community. In general, the manuscript is well written and well organized, and I recommend publication after the authors address the minor points listed below.

Specific comments/recommendations:

(1) Page 2 line 24: The authors mention the model predecessors (CAFE and UWCM), but do not differentiate the previous 0-D model (UWCM) from F0AM. At the end of this paragraph, it would be useful to briefly state the major additions or modifications that

are later described in detail.

Also, at this point or elsewhere, it would be good to reference good agreement between the UWCM and DSMACC found by Anderson et al. (2016), which could serve as further F0AM validation.

(2) Page 3 line 13: Before explaining the special option of constraining total NOX, it should be stated that the framework allows for constraining the model to concentrations of any individual chemical species specified within the chosen chemical mechanism. This may be obvious, but if the goal is to encourage use amongst those who are not familiar with modeling, it would be worthwhile to state.

Also at this point, it could be explained that observations can be used to constrain a species throughout a model step, to initialize concentrations at the beginning of each model step, or to initialize the first model step only. While the examples applications illustrate this, it might good to highlight that aspect of F0AM's flexibility here.

(3) Page 4 line 21: Two of the three F0AM photolysis methods (MCM and hybrid) are compared to TUV. My understanding is that TUV may differ from the bottom-up method (due to choices in cross section and quantum yields) and bottom-up method may differ from the hybrid method (due to interpolation across the lookup tables). In this case, for completeness, the authors should mention how bottom-up compares to TUV.

(4) Page 7 line 22: is total NOx constrained, or are NO and NO2 individually constrained?

(5) Page 7 line 26 "Chemistry is MCMv3.3.1..." –> "The chemical mechanism employed is MCMv3.3.1...", or similar phrasing

(6) Page 8 line 24: "There are significant discrepancies." –> "There are significant discrepancies at other times throughout the day", or similar additional phrasing.

(7) Page 9 line 16: "minor issues..." –> "minor discrepancies..." In other words, this analysis shows that there are differences between mechanisms, but without compari-
son to observations, no mechanism(s) can be considered more "correct".

(8) Page 10 line 4: "between the community" –> "with the community"

(9) Figure 7: x axis label –> HO2 production rates (ppt s-1)

References:

Anderson, D. C. et al. A pervasive role for biomass burning in tropical high ozone/low water structures. Nat. Commun. 7:10267 doi: 10.1038/ncomms10267 (2016).

---

## Referee Comment (RC2) · K.M Emmerson (Referee) · 29 Jul 2016

This work presents a box model environment for the testing of chemical mechanisms. F0AM represents an advancement over some of the other available mechanism testing codes in that the user has the ability to change the photolysis calculation method amongst other things. The paper takes the reader through a number of model examples which I can see would be adaptable to most experimental set-ups: a fixed location (Eulerian) setup, and a Lagrangian set-up, where the box is able to move in 3D space. Data from several field campaigns are used to demonstrate how F0AM operates in both set-ups.

The paper is a well written, enjoyable read and I recommend publication in GMD after consideration of the following points:

I found figure 2 and its associated write-up in paragraph 3 of page 4 confusing. Where it says "differences between the TUV and hybrid values for C2H5CHO and CH3COCH3….." I can't see an entry for C2H5CHO in the x axis of figure 2, but I can see two separate entries for CH3COCH3 (one next to HPALD in the upper plot, the other next to CH3CHO->CH3CO in the lower plot; only the first of these entries shows both TUV and MCM together). There are no plotted TUV values for the first 3 species listed on the lower plot (crotonaldehyde, benzaldehyde and diethylketone), and it is not clear why they have been included.

Page 5, line 17, would it be possible to include mechanisms in a kpp format in a future release?

Page 7, photochemical chamber paragraph. I've done a few chemical box model studies on photochemical chamber experiments and have found that you need some way of accounting for the initial wall loss of species when they're first injected into the chamber. Is this accounted for in the model, or does the user need to make an assumption that the injected concentration does not equal the initial concentration of reactant?

Figure 6. This is where it gets really interesting, from a mechanism point of view. The first thing to note is that you've chosen two versions of the MCM and two versions of the carbon bond mechanism, and both show an increase in OH concentrations with the evolution of the newer versions.

The second point is a request for some additional observations to be plotted. Figure 6 shows a time series representing the SENEX campaign. I'm not sure whether radical species where measured, but plotting some of the secondary species (isoprene products, formaldehyde?) would show how well the chemistry schemes are performing. This is a common criticism of mechanism comparison papers – the mechanisms are compared with each other but not to observations which would tell us which scheme performed 'better' for that particular model set-up. After all, this is what users are really after!

There are a couple of problems in the reference list where subscripting hasn't worked properly: see page 12 line 11 and also line 49.

I also wanted to have a go with the F0AM software, as its capabilities are of interest to me. I downloaded version 3.1 from the supplementary section. I used Matlab many years ago during my PhD, however I didn't find this code intuitive. What I was after was details of how to execute an example script. There are example scripts with the download, but the suggested technique in the readme.pdf is to "dive in", which is a bit daunting. It would be useful if this readme document started with a guide about how to set up the framework (windows/linux?) and run an example, as my initial thought was that it was going to take me a while to set-up properly.

I tried executing the "exampleSetup_chamber.m on linux and got the message there was an undefined function of variable F0AM_ModelCore. It took me a little while to work out that I needed to add every folder to the model pathname in order to run the script. Once I did this, I was able to run the model and out popped four figures. I had problems with a couple of the other example scripts, mainly due to licensing problems at my end I think (maximum number of statistical tool licences had been used, so the diurnal cycle script and the mechanism intercomparison script crashed). If this isn't the case then I'm happy to be contacted by the authors to get it working.

The script was very well commented, with instructions on how to change the input variables.

---

## Author Comment (AC1) · 3 Sep 2016

The Framework for 0-D Atmospheric Modeling (F0AM) v3.1, Wolfe et al., GMD (2016)

Response to Reviewer Comments

We thank the reviewers for their constructive feedback. Reviewer comments are listed below in **bold**, responses are in normal font, and updated text in *italics*.

**Anonymous Referee # 1**

**This manuscript introduces the Framework for 0-D Atmospheric Modeling (F0AM) v3.1, a MATLAB-based platform that emphasizes accessibility and flexibility. This framework has already proven to be a useful tool implemented in previous studies, and with the modifications and added functionality detailed in the manuscript, I anticipate an increase in use and development from the atmospheric modeling community. In general, the manuscript is well written and well organized, and I recommend publication after the authors address the minor points listed below.**

**Specific comments/recommendations:**
**(1) Page 2 line 24: The authors mention the model predecessors (CAFE and UWCM), but do not differentiate the previous 0-D model (UWCM) from F0AM. At the end of this paragraph, it would be useful to briefly state the major additions or modifications that are later described in detail. Also, at this point or elsewhere, it would be good to reference good agreement between the UWCM and DSMACC found by Anderson et al. (2016), which could serve as further F0AM validation.**

We have added the following text at the end of this paragraph:

*Anderson et al. (2016) found excellent agreement between UWCM and DSMACC when modelling ozone production in the tropical Western Pacific, adding some confidence to our approach. Several major changes distinguish F0AM from UWCM. While UWCM was built around the Master Chemical Mechanism (MCM), F0AM facilitates use of nearly any chemical mechanism, and a library of common mechanisms are included (Sect. 2.3). Implementing these mechanisms required significant modifications to the photolysis parameterizations, and more options for photolysis are now available (Sect. 2.2). Other new features in F0AM include an option to constrain total NOx (Sect. 2.1) and improved visualization tools.*

**(2) Page 3 line 13: Before explaining the special option of constraining total NOX, it should be stated that the framework allows for constraining the model to concentrations of any individual chemical species specified within the chosen chemical mechanism. This may be obvious, but if the goal is to encourage use amongst those who are not familiar with modeling, it would be worthwhile to state.**

**Also at this point, it could be explained that observations can be used to constrain a species throughout a model step, to initialize concentrations at the beginning of each model step, or to initialize the first model step only. While the examples applications illustrate this, it might good to highlight that aspect of F0AM's flexibility here.**

The original manuscript does state these points in the preceding paragraph, though perhaps too briefly. We added the following paragraph to this section:

*Concentrations for each chemical species within a given mechanism can be initialized and/or constrained to observations or user-specified values; the default initial concentration is 0. The way chemical constraints are handled depends on the specific scenario. Any constrained species can be held constant throughout a model step, which may be desirable when simulating diurnal cycles using discrete observations (Sect. 3.3 and 3.4). Alternatively, concentrations can be initialized at the beginning of a step and allowed to evolve over time, which may be more appropriate when modelling laboratory experiments or Lagrangian plumes (Sect. 3.1 and 3.2).*

**(3) Page 4 line 21: Two of the three F0AM photolysis methods (MCM and hybrid) are compared to TUV. My understanding is that TUV may differ from the bottom-up method (due to choices in cross section and quantum yields) and bottom-up method may differ from the hybrid method (due to interpolation across the lookup tables). In this case, for completeness, the authors should mention how bottom-up compares to TUV.**

We apologize for the confusion here. The hybrid method is essentially an extension of the bottom-up method. The main difference is that the bottom-up approach requires input of a radiation spectrum, while the hybrid method employs solar spectra calculated from TUV. Both use the same cross sections and quantum yields. Thus, the comparison between hybrid and TUV is essentially equivalent to comparing bottom-up with TUV. Interpolation errors from the use of lookup tables in the hybrid method are relatively small. The other reviewer also found this section and Figure 2 to be confusing; please see our response to her for revisions.

**(4) Page 7 line 22: is total NOx constrained, or are NO and NO2 individually constrained?**

NO and NO2 are initialized but not constrained. We have modified the text appropriately.

**(5) Page 7 line 26 "Chemistry is MCMv3.3.1. . ." –> "The chemical mechanism employed is MCMv3.3.1. . .", or similar phrasing**
**(6) Page 8 line 24: "There are significant discrepancies." –> "There are significant discrepancies at other times throughout the day", or similar additional phrasing.**
**(7) Page 9 line 16: "minor issues. . ." –> "minor discrepancies. . ." In other words, this analysis shows that there are differences between mechanisms, but without comparison to observations, no mechanism(s) can be considered more "correct".**
**(8) Page 10 line 4: "between the community" –> "with the community"**
**(9) Figure 7: x axis label –> HO2 production rates (ppt s-1)**

All above changes incorporated. Thank you.

**References:**
**Anderson, D. C. et al. A pervasive role for biomass burning in tropical high ozone/low water structures. Nat. Commun. 7:10267 doi: 10.1038/ncomms10267 (2016).**

**Kathryn Emmerson**

**This work presents a box model environment for the testing of chemical mechanisms. F0AM represents an advancement over some of the other available mechanism testing codes in that the user has the ability to change the photolysis calculation method amongst other things. The paper takes the reader through a number of model examples which I can see would be adaptable to most experimental set-ups: a fixed location (Eulerian) setup, and a Lagrangian set-up, where the box is able to move in 3D space. Data from several field campaigns are used to demonstrate how F0AM operates in both set-ups. The paper is a well written, enjoyable read and I recommend publication in GMD after consideration of the following points:**

**I found figure 2 and its associated write-up in paragraph 3 of page 4 confusing. Where it says "differences between the TUV and hybrid values for C2H5CHO and CH3COCH3. . ..." I can't see an entry for C2H5CHO in the x axis of figure 2, but I can see two separate entries for CH3COCH3 (one next to HPALD in the upper plot, the other next to CH3CHO->CH3CO in the lower plot; only the first of these entries shows both TUV and MCM together). There are no plotted TUV values for the first 3 species listed on the lower plot (crotonaldehyde, benzaldehyde and diethylketone), and it is not clear why they have been included.**

The C2H5CHO entry is in the top plot, between CH3CHO and C3H7CHO. There are multiple entries for some species because of different photolysis pathways. For some species, like benzaldehyde, J-values are not available from TUVv5.2 or the MCM – this is why the hybrid values were chosen for the denominator – but entries were included for these for completeness.

In Figure 2, entries without a corresponding symbol have been deleted. Other entries are re-organized in a more logical fashion. The description paragraph now reads as follows:

*Figure 2 compares photolysis frequencies calculated with the MCM parameterization and the F0AM hybrid method for a single set of inputs (SZA = 0°, altitude = 0.5km, albedo = 0.01, $O_3$ column = 350 DU). The overhead $O_3$ column and albedo for this comparison are chosen to optimize average agreement between the hybrid and MCM values, since the exact solar spectra underlying the MCM parameterization are not available. The two methods agree to within ±20% for inorganics, organic nitrates and some VOC. Agreement is more variable for larger VOC, in part due to varying quantum yields; for example, MCM uses different branching ratios for the glyoxal photolysis channels than those recommended by JPL or IUPAC. Figure 2 also compares hybrid values with those output directly by TUVv5.2, which includes its own photolysis algorithm. Photolysis frequencies for these two methods generally agree to within ±20%, as expected since both utilize identical solar spectra and generally comparable cross sections and quantum yields. Differences for $N_2O_5$, $CH_3CHO$ and MEK photolysis stem from the choice of quantum yields. Differences for $C_2H_5CHO$ and $CH_3COCH_3$ photolysis are due to known errors in TUVv5.2 that will be resolved in the next release (S. Madronich, personal communication, 2016). Based on the above comparison, we recommend the hybrid method over the MCM parameterization for most "real atmosphere" simulations.*

**Page 5, line 17, would it be possible to include mechanisms in a kpp format in a future release?**

Yes, one would just need a KPP-to-F0AM translator similar to that used for MCM-FACSIMILE files. Incidentally, Wolfe already has some (rough) code to do this for MECCA and would be happy to share – it is just not ready for inclusion in the model release. We have changed this sentence as follows:

*A utility is available for converting mechanisms from the FACSIMILE (MCPA Software) format into the F0AM input format, and a similar utility for converting KPP-formatted mechanisms (Damian et al., 2002) may be included in a future release.*

**Page 7, photochemical chamber paragraph. I've done a few chemical box model studies on photochemical chamber experiments and have found that you need some way of accounting for the initial wall loss of species when they're first injected into the chamber. Is this accounted for in the model, or does the user need to make an assumption that the injected concentration does not equal the initial concentration of reactant?**

Assuming the reactant concentration is measured, it would seem most appropriate to initialize the model with the measured concentrations and not a calculated injection concentration – this would inherently account for early wall passivation. For most experiments, wall losses can be accounted for using a simple $1^{st}$-order reaction with an appropriate rate constant – presumably one chosen to match concentration decays in a control experiment.  F0AM does not make special provisions for this other than allowing for the addition of sub-mechanisms. We have added the following sentence:

*We do not consider wall losses in this simple example, but such processes are typically represented with additional first-order loss reactions (Wolfe et al., 2012).*

**Figure 6. This is where it gets really interesting, from a mechanism point of view. The first thing to note is that you've chosen two versions of the MCM and two versions of the carbon bond mechanism, and both show an increase in OH concentrations with the evolution of the newer versions.**

We did not want to dive too deeply into these details since we do not have HOx observations for comparison and this is not the focus of the paper, but we agree that this is an interesting point. We have added the following text:

*Even this relatively short simulation is revealing. For example, both the MCM and carbon bond mechanisms exhibit an increase in OH and $HO_2$ between the old and new mechanism versions.*

**The second point is a request for some additional observations to be plotted. Figure 6 shows a time series representing the SENEX campaign. I'm not sure whether radical species where measured, but plotting some of the secondary species (isoprene products, formaldehyde?) would show how well the chemistry schemes are performing. This is a common criticism of mechanism comparison papers – the mechanisms are compared with each other but not to observations which would tell us which scheme performed 'better' for that particular model set-up. After all, this is what users are really after!**

We agree that comparing to observations is a far more valuable exercise for a true mechanism evaluation. We are currently preparing a manuscript that will present an inter-comparison focused on

HCHO observations during SENEX, and we would prefer to not steal the thunder from that paper if possible. We have added the following text at the end of this paragraph:

*A true mechanism evaluation also requires comparison to measurements where possible. The SENEX dataset lacks $HO_x$ observations, but it does include a wide range of isoprene oxidation products. Work is ongoing to evaluate isoprene chemistry within these mechanisms using observations of HCHO and other species from the full SENEX mission (Marvin et al., 2016).*

*Marvin, M., Wolfe, G. M., and Salawitch, R., et al.: Evaluating mechanisms for isoprene oxidation using a constrained chemical box model and SENEX observations of formaldehyde, in preparation, 2016.*

**There are a couple of problems in the reference list where subscripting hasn't worked properly: see page 12 line 11 and also line 49.**

Corrected, thank you.

**I also wanted to have a go with the F0AM software, as its capabilities are of interest to me. I downloaded version 3.1 from the supplementary section. I used Matlab many years ago during my PhD, however I didn't find this code intuitive. What I was after was details of how to execute an example script. There are example scripts with the download, but the suggested technique in the readme.pdf is to "dive in", which is a bit daunting. It would be useful if this readme document started with a guide about how to set up the framework (windows/linux?) and run an example, as my initial thought was that it was going to take me a while to set-up properly.**

**I tried executing the "exampleSetup_chamber.m on linux and got the message there was an undefined function of variable F0AM_ModelCore. It took me a little while to work out that I needed to add every folder to the model pathname in order to run the script. Once I did this, I was able to run the model and out popped four figures. I had problems with a couple of the other example scripts, mainly due to licensing problems at my end I think (maximum number of statistical tool licences had been used, so the diurnal cycle script and the mechanism intercomparison script crashed). If this isn't the case then I'm happy to be contacted by the authors to get it working.**

**The script was very well commented, with instructions on how to change the input variables.**

We are glad to hear that the reviewer took the time to look at the model and support materials, and sad to hear that she encountered issues running it.

We have now included an additional "Getting Started" document that is meant to facilitate use for those relatively unfamiliar with Matlab. We welcome further comments on how to improve model accessibility.

The problems with the stat toolbox license are likely related to our use of the nanmean and nanmedian functions. We have eliminated these function calls, so that only the Matlab built-in functions are needed.